# Self-Supported Reduced Graphene Oxide Membrane and Its Cu^2+^ Adsorption Capability

**DOI:** 10.3390/ma14010146

**Published:** 2020-12-31

**Authors:** Yangjinghua Yu, Zhong Wang, Runjun Sun, Zhihua Chen, Meicheng Liu, Xiang Zhou, Mu Yao, Guohe Wang

**Affiliations:** 1College of Textile and Clothing Engineering, Soochow University, Suzhou 215006, China; 20174015009@stu.suda.edu.cn (Y.Y.); wangzhong1215@suda.edu.cn (Z.W.); 2School of Textiles and Materials, Xi’an Polytechnic University, Xi’an 710048, China; sunrunjun@xpu.edu.cn; 3Jiangsu College of Engineering and Technology, Nantong 226014, China; czh@jcet.edu.cn (Z.C.); liumeicheng@jcet.edu.cn (M.L.); tea@jcet.edu.cn (X.Z.)

**Keywords:** graphene oxide, reduced graphene oxide, membrane, reduction, membrane adsorption

## Abstract

Graphene stratiform membrane materials have been recently applied to heavy metal removal in aqueous systems via adsorption due to their high mechanical strength, chemical stability, and other properties. We applied reduced graphene oxide (rGO) alone as an adsorbent to remove heavy metal ions from wastewater. Self-supported rGO membrane was prepared using a green reduction method with sodium hydrosulfite. We used the Raman spectra to observe the structure of the rGO membrane. The morphology of the self-supported membrane was measured by a scanning electron microscope. The Cu^2+^ adsorption performance was measured in terms of pH, reaction time, metal ion concentration, and temperature. The maximum Cu^2+^ adsorption capacity of the rGO membrane was found to be 149.25 mg/g. The adsorption process followed a pseudo-second-order kinetic model, and adsorption isotherms were simulated by the Freundlich model.

## 1. Introduction

Heavy metals are mainly found in industrial wastewater. Determining the effluent status has become increasingly complex due to industrial development. To mitigate negative impacts on the environment, the emissions of toxic substances into natural water must be controlled [1]. However, heavy metals have low biodegradability and remain stable in the environment, which causes significant harm to human health and the environment [2,3]. Copper and its derivatives, which could be considered as the most common heavy metals in wastewater, are widely used in plating baths, pigments, and fertilizers. The accumulation of Cu^2+^ in the human body could cause skin, pancreatic, heart, and brain diseases, so the removal of copper and its derivatives is vital. Effluent treatment methods for copper include electrochemical, membrane filtration, chemical oxidation, photocatalytic degradation, and adsorption [4]. Of these techniques, adsorption technology is considered the most effective as well as being low cost and practical [5].

Graphene has been widely used in various fields due to its the excellent conductivity, thermal conductivity, transmittance, and adsorption, which are attributed to its unique two-dimensional single-carbon atomic layer nanostructure [6,7]. As new-generation adsorbents, graphene-based materials have been investigated for the treatment of water in the environment due to their high adsorption affinity, open-up layer morphology, and high hydrophobic surface area [8]. Membrane technologies play an important role in environmental protection [9,10,11]. Single-atom-thick graphene membranes were confirmed to be impermeable to atoms [12]. Graphene oxide (GO), a graphene derivative, acts as a bridge for the preparation of graphene films [13,14,15]. Graphene-based films are generally built by thickness reduction passages, which could provide the mass transfer needed to maximize permeability [16].

Reduced graphene oxide (rGO) is usually achieved using chemical reducing agents (such as amino acids, oxalic acid, D-glucose, and tea polyphenols), heat treatment, and microwave and electrochemical methods [17]. rGO has not been used alone as an adsorbent to remove heavy metal ions from polluted water [18].

The purpose of this work was to study the Cu^2+^ adsorption properties of a self-supported rGO membrane in an aqueous system. The green reduction ability of the GO membrane and the optimum conditions of Cu^2+^ adsorption were determined by experiments. Cu^2+^ adsorption was analyzed using kinetics, isotherms, and thermodynamics on the rGO membrane. It was found that rGO membrane can be used to remove Cu^2+^ in polluted water.

## 2. Experimental Section

### 2.1. Materials

Graphite, sodium nitrate, and copper sulfate pentahydrate were purchased from Aladdin (Shanghai, China). Sulfuric acid (98%) and hydrochloric acid (36%) were supplied by Qiangsheng Functional Chemistry Co. Ltd. (Suzhou, China). Potassium permanganate (analytical reagent (AR)) was purchased from Chinese Medicine Group Chemical Reagent Co. Ltd. (Shanghai, China). Hydrogen peroxide (AR) was supplied by Lingfeng Chemical Reagent Co. Ltd. (Shanghai, China). Sodium hydroxide (guarantee reagent) was purchased from Yonghua Chemical Science and Technology Co. Ltd. (Suzhou, China). Sodium hydrosulfite was from Adamas (Shanghai, China).

### 2.2. GO Synthesis

The modified Hummer’s method was used for synthesis of graphene oxide by graphite [19,20,21]. The whole system was stirred throughout the preparation process. First, 46 mL concentrated sulfuric acid was placed in a beaker. In an ice bath, 1 g of graphite and 1 g of sodium nitrate were added to the concentrated sulfuric acid, and 6 g of potassium permanganate was gradually added. After 30 min, the system was heated to 35 °C for 1 h. Subsequently, 80 mL of deionized water was added and heated to 95 °C for 30 min. Then, 12 mL of 30% hydrogen peroxide solution and 200 mL of deionized water was added. After washing with 5% hydrochloric acid once or twice, the mixture was rinsed with deionized water. After centrifuging, the GO was dried at 50 °C for 24 h in a blast dryer.

### 2.3. Preparation of the rGO Membrane

Adequate dispersal of the GO solution was achieved using 0.25 g of dried GO powder and 500 mL ultrasonic distilled water. This homogeneous solution was filtered using a vacuum filter with cellulose acetate membrane. The membrane was peeled after desiccation for one night at room temperature. The mass of the prepared GO membrane was 7.96 g·m^−2^. The GO membrane was reduced with sodium hydrosulfite as the reductant [20,21,22]. After studying the reduction conditions, each sheet of GO membrane was immersed in 100 mL of 40 g·L^−1^ Na_2_S_2_O_4_ aqueous solution. The mixture was maintained at 80 °C for 50 min. To remove excessive reducing agent, the membrane was washed with deionized water several times. Finally, the rGO membrane was dried at 50 °C for 120 min.

### 2.4. Raman Spectrum Analysis

The Raman spectra (LabSpec 6, HORIBA XploRA, Lille, France) were captured with a laser light wavelength of 532 nm incident in the range of 1000–3000 cm^−1^. The sample was used without any other treatment.

### 2.5. Morphology Analysis

The morphology of the rGO membrane was observed by a scanning electron microscope (SEM; FESEM, S-4800, Hitachi, Tokyo, Japan) after spraying the sample with gold. The thickness of the rGO film was measured by a O6223-01 digital outside micrometer (Aladdin Industrial Corporation, Shanghai, China).

### 2.6. Electrical Surface Resistance Analysis

The electrical surface resistance of the membrane was measured using a standard four-probe method using a digital four-probe tester.

### 2.7. Adsorption Experiment

A standard solution of 1 g·L^−1^ Cu^2+^ was prepared using copper sulfate pentahydrate. The pH value was adjusted with sodium hydroxide and hydrochloric acid. The iodine flasks containing 25 mL solution were shaken at a rotational speed of 120 rpm using an incubator shaker in the Cu^2+^ batch adsorption experiments.

Next, 6 mg of adsorption membrane was added to 100 mg·L^−1^ (ppm) Cu^2+^ solution for different durations (10, 20, 40, 60, 120, 180, 240, 300, 360, 420, and 480 min) at an initial pH of 6 and constant temperature of 25 °C. This adsorption membrane was shaken in different initial concentrations of Cu^2+^ solution (50, 75, 100, and 125 ppm) for 10, 20, 40, 60, 120, 180, and 240 min with the temperature maintained at 25 °C.

The initial pH value of the 100 ppm Cu^2+^ solution was adjusted from 2 to 6. Then, 6 mg film was used for adsorption for 180 min at 25 °C. The 6 mg adsorption membrane was shaken in different initial concentrations of Cu^2+^ solutions (50, 75, 100, 125, and 150 ppm) at different temperatures (25, 30, and 35 °C) at pH 6 for 240 min to obtain the isotherms.

The Cu^2+^ concentration was determined using inductively coupled plasma emission spectrum (ICP-OES, Icap6300, Waltham, MA, USA). The ICP-OES result provided the concentration of the residual Cu^2+^ in the solution after adsorption. The adsorption capacity of Cu^2+^ adsorption (*q_e_*) and the percentage of removal (*R_e_*) were calculated as follows:(1)$$q_{e}=\frac{(C_{0}-C_{e})V}{W}$$
(2)$$R_{e}=\frac{C_{0}-C_{e}}{C_{0}}\times 100\%$$
where the initial and equilibrium concentration of Cu^2+^ are expressed by *C*_0_ and *C_e_* (mg∙L^−1^), respectively; the volume of the Cu^2+^ aqueous solution and the weight of the adsorbent membrane are indicated by *V* (L) and *W* (g), respectively.

## 3. Results and Discussion

### 3.1. Reduction of GO Membrane

Many methods have been used to reduce graphene oxide, amongst which Na_2_S_2_O_4_ could be regarded as a better reducing agent in chemical reduction methods [23]. Considering its low price and environmental friendliness, Na_2_S_2_O_4_ has the potential to effectively manufacture rGO membranes.

The effect of Na_2_S_2_O_4_ concentration on electrical surface resistance was investigated (Table 1). The low-concentration Na_2_S_2_O_4_ (10 g·L^−1^) solution could not provide higher conductivity because it did not sufficiently lessen the oxygen-containing groups of GO to convert them into complete graphene. The electrical surface resistance of the rGO membrane constantly decreased from 480 to 370 Ω·cm^−1^ with increasing Na2S_2_O_4_ concentration from 20 to 40 g·L^−1^. However, the electrical surface resistance of the membrane increased at a Na_2_S_2_O_4_ concentration of 50 g·L^−1^. The smallest value of surface resistance was observed at 40 g·L^−1^.

Electrical surface resistance was studied with different reduction times. The data indicated that the electrical surface resistance decreased from 798.2 to 413.1 Ω·cm^−1^ (conductivity increased) with increasing reaction time from 30 to 50 min (Figure 1a). With a reduction time from 60 to 70 min, the electrical surface resistance of the rGO membrane slightly increased from 472.6 to 501.4 Ω·cm^−1^, which may be due to over-reduction. With prolonging reduction time from 60 to 70 min, the reduction induced defects on the rGO membrane and inhibited charge transfer. Complete reduction of the GO membrane should be performed at 50 min to convert into an rGO membrane.

The influence of reducing temperature on conductivity was investigated (Figure 1b). The electrical surface resistance of the membrane constantly decreased from 299.83 to 147.98 Ω·cm^−1^ with increasing reaction temperature from 50 to 80 °C. However, the electrical surface resistance grew slightly due to over-reduction when the temperature was 90 °C. Therefore, the highest conductivity response was observed at 80 °C. The optimum conditions of reduction were 50 min, 80 °C, and 40 g·L^−1^. The optimum conditions were sufficient for complete reduction of the GO membrane.

### 3.2. Characterization of the rGO Membrane

Self-supported rGO membrane was prepared by the reduction of GO film (Figure 2a). The rGO membrane was black and shiny. The SEM image showed the cross section of the rGO membrane and illustrated the interlayer space structure of the rGO membrane (Figure 2b). After reduction of the GO film, the mass of the rGO membrane was 6 mg and the thickness was 4.4 µm, which was indirectly proven by the cross-sectional image of the rGO membrane. Its flexibility is shown in Figure 2c.

The Raman spectra of the rGO membrane was characterized by a G band at 1566 cm^−1^ and a D band at 1335 cm^−1^ (Figure 3). The important characteristic G peak of graphene was caused by the in-plane vibration of sp2 carbon atoms. Figure 3 shows the existence of rGO due to the overall stronger absorption intensity of rGO. The D or G peak showed stronger intensity on the rGO membrane. The strength ratio (I_D_/I_G_) of the D and G peaks indirectly illustrated the disorder and defects of folds, edges, and pores. The D peak being stronger than the G peak indicated that the rGO membrane produced structure defects during the reduction process. Therefore, the reduction method is in agreement with the results of other common effective reduction methods.

### 3.3. Cu^2+^ Adsorption Performance of the rGO Membrane

#### 3.3.1. Effect of Contact Time

The effect of contact time on the Cu^2+^ adsorption of the rGO membrane is shown at 100 ppm in Figure 4a. The highest Cu^2+^ adsorption rate appeared in the first 60 min. The amount of Cu^2+^ adsorbed onto the rGO membrane linearly increased with an increase in contact time, which might be due to the strong influence of the charges on membrane selectivity [20]. After 60 min, the adsorption quantity of Cu^2+^ grew briefly, and then the adsorption capacity increased smoothly with increasing time. The adsorption amount did not significantly increase following further increase in contact time, which might be the reason for the decrease in the driving force. Through the above analysis, the optimal balance time was chosen as 240 min for subsequent experiments. The maximum Cu^2+^ adsorption capacity of the rGO membrane was 149.25 mg/g; compared to the adsorption capacity of other adsorbents [24,25,26,27], the capacity of the rGO prepared in this work was greater (Table 2). The promotion of adsorption could be attributed to the layer structure of the rGO film, which provided an inner layer pass road. Therefore, Cu^2+^ ions could effectively pass through the layers and adsorbed by the surface of the rGO.

The removal of Cu^2+^ by the membrane was investigated under four initial concentrations, and the results are depicted in Figure 4b. The adsorption rate was exceedingly high in the first 20 min, and the same trend in the adsorption properties with different initial concentrations was found with the change in contact time. The adsorption properties were also promoted with increasing initial concentration.

#### 3.3.2. Effect of Initial pH

Figure 5 demonstrates the impact of pH on the adsorption capacity and removal of Cu^2+^ of the rGO membrane. To avoiding precipitation, when pH was changed from 2 to 6, the adsorption of Cu^2+^ increased from 17.67 to 119.67 mg·g^−1^. The pH strongly affected Cu^2+^ adsorption by the rGO membrane. As the pH level increased, the removal efficiency increased from 3.59% to 24.37%. The adsorption capacity was maximum at pH 6, which might be due to the electronegativity of graphene in water, which can be absorbed by the action of positive and negative charges.

#### 3.3.3. Adsorption Kinetic Study

Cu^2+^ adsorption kinetic models were analyzed to investigate the mechanism of the rGO membrane adsorption process. Changes in adsorption with time were quantified using a suitable kinetic model. The mechanism was interpreted by comparing the ability of the pseudo-first-order and pseudo-second-order models to fit the experimental data. The equations of the pseudo-first-order and pseudo-second-order kinetic models are expressed as follows [28,29]:(3)$$\log(q_{e}-q_{t})=\log q_{e}-\frac{k_{1}}{2.303}t$$
(4)$$\frac{t}{q_{t}}=\frac{1}{k_{2}q_{e}^{2}}+\frac{1}{q_{e}}t$$
where *q_e_* and *q_t_* are the amount of Cu^2+^ adsorbed on the rGO membrane at equilibrium time and contact time (mg·g^−1^), respectively; *k*_1_ (min^−1^) and *k*_2_ (g (mg min)^−1^) are the rate constants of the pseudo-first-order and pseudo-second-order models, respectively.

The fitted curves for the pseudo-first-order and pseudo-second-order kinetic models are presented in Figure 6. The kinetic parameters are listed in Table 3 for Cu^2+^ adsorption on the rGO membrane at four initial concentrations. The lowest coefficient obtained by the pseudo-second-order model was 0.9929, which was better than the coefficient of the pseudo-first-order model. The experimental *q_e_* value was close to the calculated *q_e_*_2_ in the pseudo-second-order model, but *q_e_* was significantly different from the calculated *q_e_*_1_ value in the pseudo-first-order model. Therefore, the adsorption process was more accurately captured by the pseudo-second-order model. The adsorption process was mainly chemisorption.

Hydrolysis, flocculation, and ion exchange may lead to the adsorption of heavy metals by carbon-based materials, that is, the adsorption of Cu^2+^ onto the rGO membrane is the result of the chemical reaction between the negative charge on the membrane surface and the positive metal ions as well as the electrostatic interaction. The graphene surface is negatively charged, which enables efficient removal of cationic contaminants [30]. To further study the internal action during the adsorption process, the intraparticle diffusion model [31] was applied:(5)qt=kpit12+ci
where *k_pi_* is the rate coefficient of intraparticle diffusion (mg·g^−1^·min^−1/2^), and *c_i_* is the thickness of the boundary layer that is constantly affected. The values of *k_pi_* and *c_i_* are calculated by the slope and intercept of *q_t_* vs. *t*^1/2^, respectively. The larger the *c_i_* value, the greater the boundary layer effect.

The mass transfer action of Cu^2+^ adsorption was examined using the intraparticle diffusion model on the rGO membrane. Figure 7 plots *q_t_* vs. *t*^1/2^ at four initial concentrations. The parameters are listed in Table 4 for intraparticle diffusion. Two related lines fitted by data points describe the external diffusion and intraparticle diffusion in the plot of *q_t_* vs. *t*^1/2^. For the four different initial Cu^2+^ concentrations, *k_p_*_2_ was lower than *k_p_*_1_, and *c*_1_ was smaller than *c*_2_, which illustrated that the removal rate of Cu^2+^ was faster in the beginning. The first process occurred during the first 40 min. The instantaneous diffusion stage was indicated by the large amount of Cu^2+^ being rapidly adsorbed by the rGO membrane. The second process occurred during 80–240 min, where the rate of adsorption was controlled by intraparticle diffusion. The intraparticle diffusion rate gradually decreased and gradually attained equilibrium due to the accumulation of Cu^2+^ adsorbed on the exterior surface.

#### 3.3.4. Adsorption Thermodynamic Study

The change in temperature is important in the adsorption process. Figure 8a shows the effect of initial concentration in the range of 50 to 150 ppm on Cu^2+^ adsorption at three different temperatures. The adsorption capacity of Cu^2+^ increased with the increase in initial concentration, which might be due to the diffusion rate of Cu^2+^ increasing with temperature.

The models of adsorption isotherm were used to depict the reactivity of the adsorbate and adsorbent during the adsorption process. The Freundlich and Temkin isotherm models were applied to fit the experimental data at three different temperatures. The Freundlich isotherm is often used assuming heterogeneous multilayer adsorption. The Freundlich model is generally indicated as follows [32]:(6)$$\ln q_{e}=\frac{1}{n}\ln C_{e}+\ln K_{F}$$
where *K_F_* and *n* are the Freundlich constants of the adsorption capacity and intensity, respectively. The adsorption capacity and the concentration of heavy metal are *q_e_* (mg·g^−1^) and *C_e_* (mg·L^−1^) in solutions, respectively. When the value of 1/*n* is less than 1, the Langmuir isotherm hypothesis is tenable; conversely, cooperative action is indicated in adsorption.

The Temkin isotherm assumes that binding energy is uniformly distributed at maximum binding energy during adsorption. The heat of adsorption decreases linearly due to adsorbent−adsorbate interactions with coverage of all the molecules in the layer in this model [33,34]. The Temkin model is commonly expressed as follows:(7)$$q_{e}=B\ln K_{T}+B\ln C_{e}$$
where *B* = *RT*/*b*; *R* (8.314 J·mol^−1^·K^−1^) is the universal gas constant; the absolute temperature and the Temkin isotherm constant are expressed by *T* (K) and *b*, respectively; *B* and the heat of adsorption are related. The maximum binding energy is represented by *K_T_* in the equilibrium binding constant.

The fitted curves are presented in Figure 8b,c for the two isotherm models, the parameters of which are listed in Table 5. The value of 1/*n* was greater than 1 in the Freundlich model coefficient, indicating cooperative adsorption for the multilayer adsorption process. The adsorption of the experiment conformed to the Freundlich adsorption isotherm model. Comparing the regression coefficient (*R*^2^) values of Freundlich and Temkin adsorption isotherms, the Freundlich adsorption isotherm more accurately reflected the Cu^2+^ adsorption process on the rGO membrane. Therefore, the Freundlich adsorption isotherm was more suitable. The Cu^2+^ adsorption process on the rGO membrane involved diffusion and intraparticle diffusion per our thermodynamics research.

## 4. Conclusions

Self-supported rGO membrane with interlayer space structure was prepared using a green reduction method with sodium hydrosulfite. The optimum reduction conditions were 50 min, 80 °C, and 40 g·L^−1^ reduction solution. The Raman spectra proved the existence of rGO, and the SEM image illustrated the structure of the interlayer spaces. The Cu^2+^ adsorption on rGO membrane was strongly dependent on pH and time. Kinetic research of Cu^2+^ agreed with the pseudo-second-order kinetic model, and the adsorption isotherm was consistent with the Freundlich model. The rGO membrane could effectively remove Cu^2+^ in aqueous solution. The adsorption mechanism was chemisorption. Therefore, self-supported rGO film shows promise as an adsorption material for the removal of heavy metal ions for the treatment of polluted water due to its easy synthesis, low cost, and convenient separation, playing an important role in environmental remediation.

## Figures and Tables

**Figure 1 materials-14-00146-f001:**
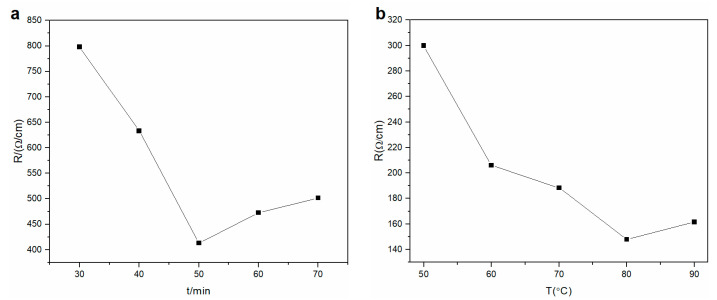
Effect of (**a**) reaction time of reduction agent (50 °C, 40 g·L^−1^) and (**b**) reduction temperature (concentration = 40 g·L^−1^; reduction time = 50 min) on electrical surface resistance of the rGO membrane.

**Figure 2 materials-14-00146-f002:**
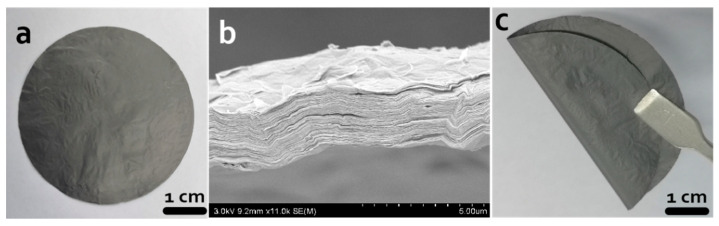
Morphology pictures. (**a**,**c**) Digital images of the self-supported rGO membrane; (**b**) SEM image of the cross section.

**Figure 3 materials-14-00146-f003:**
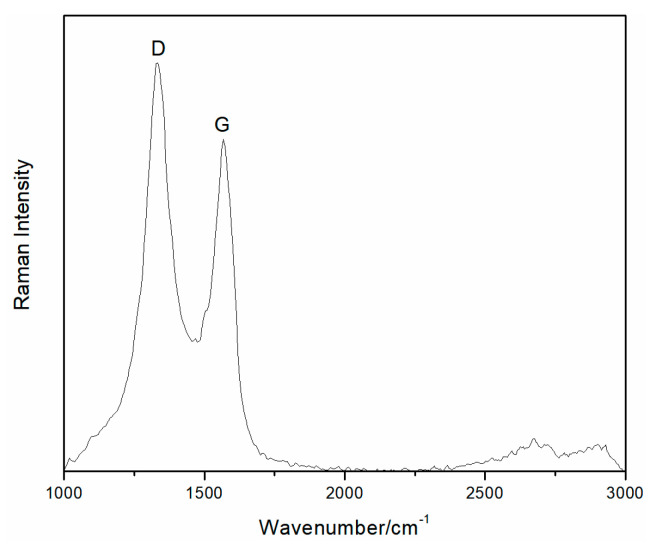
Raman spectra of the rGO membrane.

**Figure 4 materials-14-00146-f004:**
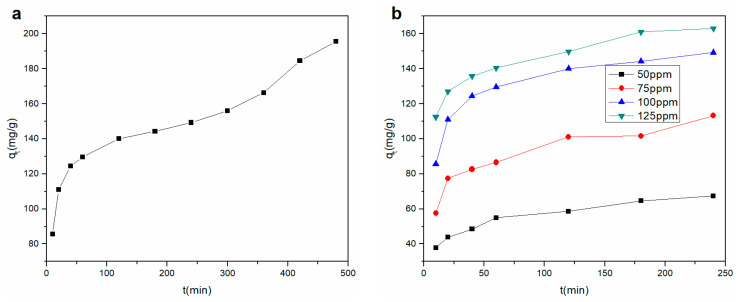
Effect of contact time for (**a**) Cu^2+^ adsorption onto the rGO membrane (**b**) with four different concentrations.

**Figure 5 materials-14-00146-f005:**
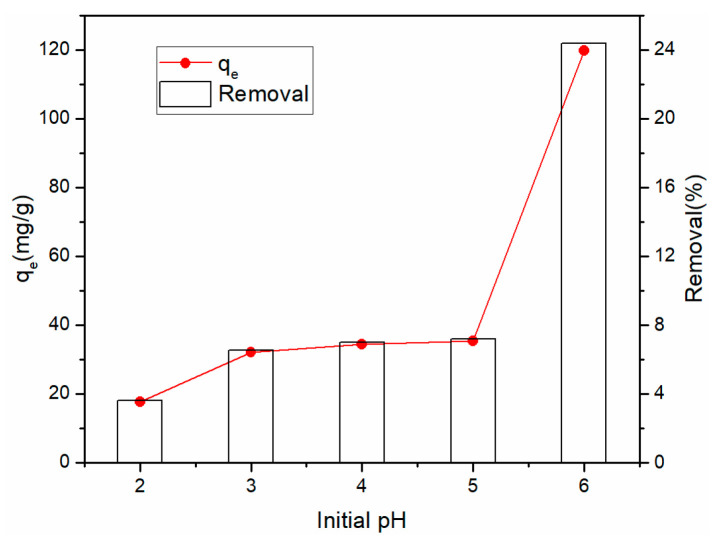
Effect of pH on the removal and adsorption of Cu^2+^.

**Figure 6 materials-14-00146-f006:**
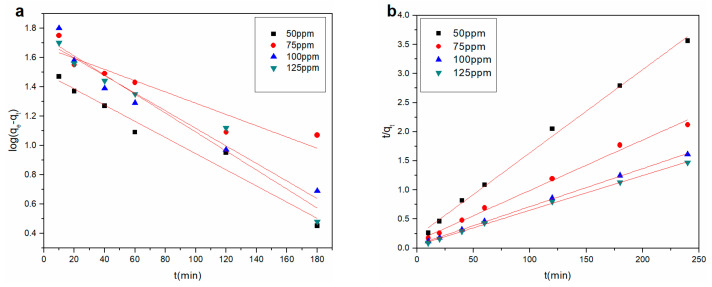
(**a**) Pseudo-first-order and (**b**) pseudo-second-order kinetic model for Cu^2+^ adsorption onto the rGO membrane.

**Figure 7 materials-14-00146-f007:**
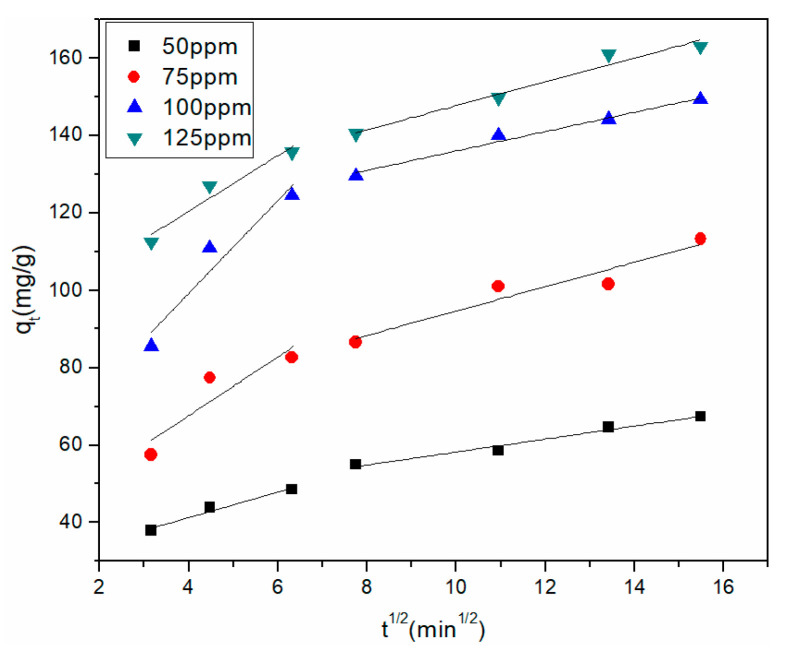
Intraparticle diffusion model of Cu^2+^ adsorption by the rGO membrane.

**Figure 8 materials-14-00146-f008:**
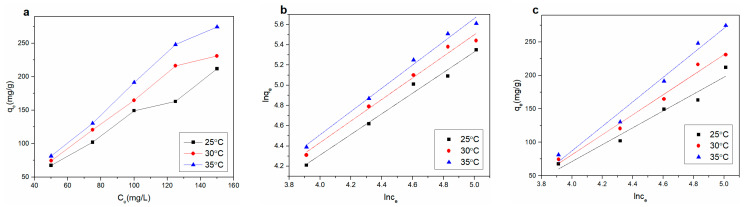
(**a**) Adsorption isotherms of Cu^2+^ on rGO membrane at different temperatures; (**b**) Freundlich and (**c**) Temkin isotherms for adsorption of Cu^2+^.

**Table 1 materials-14-00146-t001:** The effect of different concentrations of Na_2_S_2_O_4_ solution on the electrical surface resistance of the reduced graphene oxide (rGO) membrane (temperature = 50 °C; reduction time = 50 min).

Concentration of Na_2_S_2_O_4_ (g∙L^−1^)	Electrical Surface Resistance (Ω∙cm^−1^)
10	5 × 10^4^
20	480
30	410
40	370
50	560

**Table 2 materials-14-00146-t002:** Comparison of Cu^2+^ adsorption of rGO with other adsorbents.

Adsorbent	Adsorption Capacity (mg·g^−1^)	Adsorption Conditions			Reference
		pH	Time (h)	*C*_0_ (ppm)	
N-rGO	91	7	9	100	[24]
A-rGO	74				
J-rGO	93				
rGO-PDTC/Fe_3_O_4_	113.64	6	6	100	[25]
MnO_2_ nanotubes@rGO	121.5	5	12	200	[26]
ZnO nanorod-rGO	67.39	6	2	10	[27]
rGO	149.25	6	4	100	this work

**Table 3 materials-14-00146-t003:** Kinetic parameters for adsorption of Cu^2+^ onto the rGO membrane. *q_e_* (mg·g^−1^) and *q_t_* (mg·g^−1^) are the amount of Cu^2+^ adsorbed on the rGO membrane at equilibrium time and contact time, respectively; *k*_1_ (min^−1^) and *k*_2_ (g (mg min)^−1^) are the rate constants of the pseudo-first-order and pseudo-second-order models, respectively.

Kinetic Model	Parameter	50 ppm	75 ppm	100 ppm	125 ppm
Pseudo-first-order	*q_e_*_1_ (mg·g^−1^)	31.31	46.98	52.06	54.93
	*k*_1_ (min^−1^)	1.27 × 10^−2^	0.88 × 10^−2^	1.38 × 10^−2^	1.49 × 10^−2^
	*R* ^2^	0.9579	0.8635	0.9443	0.9494
Pseudo-second-order	*q_e_*_2_ (mg·g^−1^)	69.98	115.47	153.14	166.94
	*k*_2_ (g·mg^−1^·min^−1^)	0.98 × 10^−3^	0.60 × 10^−3^	0.71 × 10^−3^	0.70 × 10^−3^
	*R* ^2^	0.9957	0.9929	0.9994	0.9984
*q_e_* (mg·g^−1^)		67.43	113.25	149.25	163.00

**Table 4 materials-14-00146-t004:** Fitting parameters of intraparticle diffusion model of Cu^2+^ adsorption.

Concentration (ppm)	Parameters					
	Stage 1			Stage 2		
	*k_p_* _1_	*c* _1_	(*R*_1_)^2^	*k_p_* _2_	*c* _2_	(*R*_2_)^2^
50	3.30	28.04	0.9430	1.68	41.44	0.9649
75	7.60	37.18	0.6641	3.14	63.18	0.8781
100	11.98	51.28	0.8542	2.49	111.02	0.9736
125	7.18	91.68	0.8868	3.09	116.76	0.9496

**Table 5 materials-14-00146-t005:** Constants of two isotherm models for Cu^2+^ adsorption by the rGO membrane.

Isotherm Model	Constants	25 °C	30 °C	35 °C
Freundlich	*k_F_* (mg·g^−1^ (L·mg^−1^)^−1/n^)	1.69	1.46	0.82
	1/*n*	1.02	1.06	1.15
	*R* ^2^	0.9809	0.9834	0.9887
Temkin	*K_T_* (L·mg^−1^)	0.0322	0.0316	0.0295
	*B*	126.29	149.51	183.31
	*R* ^2^	0.9494	0.9778	0.9762

## Data Availability

No new data were created or analyzed in this study. Data sharing is not applicable to this article.

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
