# Peer review of "Self-Supported Reduced Graphene Oxide Membrane and Its Cu2+ Adsorption Capability"

_materials, 2020, doi:10.3390/ma14010146_

Round 1
Reviewer 1 Report
In the manuscript by Yu et al, entitled “Research on the self-support reduction graphene oxide membrane on its adsorption capacity for Cu2+”, the authors investigate the properties of reduced graphene oxide membranes to adsorb and remove heavy metals from aqueous solutions. The topic is very modern and popular nowadays as the pollution of the environment with heavy metals is a constant burden. The search for novel technologies guaranteeing adequate clearance of the heavy metals is a prerequisite for environmental science. Adsorption is considered the most successful tactic for heavy metals clearance. Nanotechnologies gain popularity in this with graphene oxide being on the frontline.
Therefore, the results in the current manuscript are of high interest.
Here, I list my concerns regarding the results and their representation.
The work needs extensive English revision. There are many places in the work that require revision and edit.
The Results and Discussion section needs more discussion on the obtained results. The authors have to discuss their results in the light of other authors’ results and regarding other methodology used for clearance of heavy metals.
The Conclusions part need edit. At its current version, it is a short resume of the obtained results. For the readers, it will be more useful if the authors draw the line of the future perspective of the work.
The authors need to discuss why they choose copper as a heavy metal to deal with. Moreover, to prove selectivity toward Cu2+ adsorption the authors need to perform a comparison with the adsorption rate of the tested RGO membranes and other heavy metals.
The authors use “reduction graphene oxide” inappropriately. The correct use is reduced graphene oxide membrane.
There are typos and grammar errors. Please, edit accordingly.
Ex. Line 66 “was used to synthesis”, correct with “was used for synthesis” and others.
The whole text contains many gramatical, stylistic and linguistic errors.
Author Response
Q1: The work needs extensive English revision. There are many places in the work that require revision and edit.Answer: We have revised the manuscript with MDPI English editing service.
Q2: The Results and Discussion section needs more discussion on the obtained results. The authors have to discuss their results in the light of other authors’ results and regarding other methodology used for clearance of heavy metals.Answer: We have added other authors’ results in Table 2. Relevant content was changed to “The maximum Cu2+ adsorption capacity of the RGO membrane was 149.25 mg/g; in comparison with the adsorption capacity of other adsorbents [23-26], that of the RGO prepared by the reduction method is greater(Table 2).”(Page 6 line 183-185).
Q3: The Conclusions part need edit. At its current version, it is a short resume of the obtained results. For the readers, it will be more useful if the authors draw the line of the future perspective of the work. Answer: We have the improper parts revised accordingly. In the revised paper, the sentence “Therefore, the RGO membrane could have important potential as an economical self-support film and efficient adsorbent to remove heavy metal ions in polluted water.” was changed to “Therefore, self-supported RGO film shows promise an as adsorption material for the removal of heavy metal ions for the treatment polluted water due to its easy synthesis, low cost, and convenient separation, playing an important role in environmental remediation.” ( Page 11 line 304-307).
Q4: The authors need to discuss why they choose copper as a heavy metal to deal with. Moreover, to prove selectivity toward Cu2+ adsorption the authors need to perform a comparison with the adsorption rate of the tested RGO membranes and other heavy metals. Answer: The reason of choosing copper have been discussed in Page 2 line 47-50. “Reduced graphene oxide is usually used chemical reducing agent (such as Amino acids, the use of oxalic acid and D-glucose and tea polyphenols), heat treatment, microwave and electrochemical methods to reduce graphene oxide. RGO has not been used alone as an adsorbent to remove heavy metal ions from polluted water[17]”had been added。It is very important to discuss other heavy metals with the RGO membrane. We would like to discuss other heavy metals and complex adsorption conditions with several kinds of heavy metals in the future experiments.
Q5: The authors use “reduction graphene oxide” inappropriately. The correct use is reduced graphene oxide membrane. Answer: We’ve recognized that the description was not accurate, so all the “reduction graphene oxide” have been corrected.
Q6: There are typos and grammar errors. Please, edit accordingly.Ex. Line 66 “was used to synthesis”, correct with “was used for synthesis” and others.Answer: We have followed all of the corrections.
Q7: The whole text contains many grammatical, stylistic and linguistic errors.Answer: We have revised the manuscript with MDPI English editing service.
Reviewer 2 Report
- Language of the manuscript should be polished.
- Include some experimental results in the abstract.
- Add major research findings in the conclusion section.
- In Raman spectra please label the peaks as D and G or G* and explain that which one high and why
- What is the novelty of the work compared to existing work? Include in the experimental section.
- Discuss the environmental impact and future prospects of the work.
Author Response
Q1: Language of the manuscript should be polished.Answer: We have revised the manuscript with MDPI English editing service.
Q2: Include some experimental results in the abstract.Answer: The abstract part has been rewritten with some experimental results.
Q3: Add major research findings in the conclusion section.Answer: We have followed the suggestion and the major research findings have been shown in Page 11 line 299-310.
Q4: In Raman spectra please label the peaks as D and G or G* and explain that which one high and whyAnswer: The Figure 3 has been corrected. In the revised paper, the sentence “So the reduction method is same with the other common effective reduction results.” was changed to “The strength ratio (ID/IG) of the D and G peaks indirectly illustrates the disorder and defects of folds, edges and pores. The D peak being stronger than the G peak indicated that the RGO membrane produced structure defects during the reduction process. So the reduction method agrees with the other common effective reduction results.”(Page 5 line 167-170).
Q5: What is the novelty of the work compared to existing work? Include in the experimental section.Answer: The preparation of reduced graphene oxide membranes with sodium hydrosulfite as reducing agent was an new environment-friendly method and the adsorption property of RGO membrane have not been discussed by other paper. To reflect the novelty, we have added other authors’ results in Table 2 and the Introduction part was added content in Page 2 line 45-48.
Q6: Discuss the environmental impact and future prospects of the work. Answer: The Conclusions part has been revised and the future prospects of the work have been given. (Page 11 line 304-307)
Round 2
Reviewer 1 Report
Reviewer:
Minor editing needed:
An example: Line 31
“Common copper in daily life is harmful. Copper is highly toxicity to humans, animals and 31 plants. Industrial wastewater can easily cause copper pollution. Effluent ..”
Please revise adequately!!!
Author Response
Dear reviewer;
Thank you for your kindly suggestion.
We had changed the part of page 1 from line 31 to 36. It had been change to “Copper and its derivatives, which could be considered as the most common heavy metals in the wastewater, are widely used in planting baths, pigment industry and fertilizer industry. It could cause skin, pancreas, heart and brain diseases with the accumulation of Cu2+ in human body, so the removing of copper and its derivatives imminently” instead of “Common copper in daily life is harmful. Copper is highly toxicity to humans, animals and plants. Industrial wastewater can easily cause copper pollution.”
And some other mistake had been correct as line 38, 92-93, 1270128.
Best wishes
yours
Guohe Wang
This manuscript is a resubmission of an earlier submission. The following is a list of the peer review reports and author responses from that submission.